# Kernel of CycleGAN as a Principle homogeneous space

**Nikita Moriakov**
Radiology, Nuclear Medicine and Anatomy
Radboud University Medical Center
nikita.moriakov@radboudumc.nl

**Jonas Adler**
Department of Mathematics
KTH – Royal Institute of Technology
Research and Physics
Elekta
jonasadl@kth.se

**Jonas Teuwen**
Radiology, Nuclear Medicine and Anatomy
Radboud University Medical Center
Department of Radiation Oncology
Netherlands Cancer Institute
jonas.teuwen@radboudumc.nl

## Abstract

Unpaired image-to-image translation has attracted significant interest due to the invention of CycleGAN, a method which utilizes a combination of adversarial and cycle consistency losses to avoid the need for paired data. It is known that the CycleGAN problem might admit multiple solutions, and our goal in this paper is to analyze the space of exact solutions and to give perturbation bounds for approximate solutions. We show theoretically that the exact solution space is invariant with respect to automorphisms of the underlying probability spaces, and, furthermore, that the group of automorphisms acts freely and transitively on the space of exact solutions. We examine the case of zero 'pure' CycleGAN loss first in its generality, and, subsequently, expand our analysis to approximate solutions for 'extended' CycleGAN loss where identity loss term is included. In order to demonstrate that these results are applicable, we show that under mild conditions nontrivial smooth automorphisms exist. Furthermore, we provide empirical evidence that neural networks can learn these automorphisms with unexpected and unwanted results. We conclude that finding optimal solutions to the CycleGAN loss does not necessarily lead to the envisioned result in image-to-image translation tasks and that underlying hidden symmetries can render the result utterly useless.

## 1 Introduction

Machine learning methods for image-to-image translation are widely studied and have applications in several fields. In medical imaging, the CycleGAN has found an important application for translating one modality to another, for instance in MR to CT translation (Han, 2017; Sjölund et al., 2015; Wolterink et al., 2017). Classically, these methods are trained in a supervised setting making their applications limited due to the a lack of good paired data. Similar issues appear in e.g. transferring the style of one artist to another (Gatys et al., 2015) or adding snow to sunny California streets (Liu et al., 2017). Unpaired image-to-image translation models such as CycleGAN (Zhu et al., 2017) promise to solve this issue by only enforcing a relationship on a distribution level, thus removing the need for paired data. However, given their widespread use, it is paramount to gain more understanding of their dynamics, to prevent unexpected things from happening, e.g., (Cohen et al., 2018). As a step in that direction, we explore the solution space of the CycleGAN in the subsequent sections of this paper.

The general task of unpaired domain translation can be informally described as follows: given two probability spaces X and Y which represent our domains, we seek to learn a mapping $G : X \to Y$

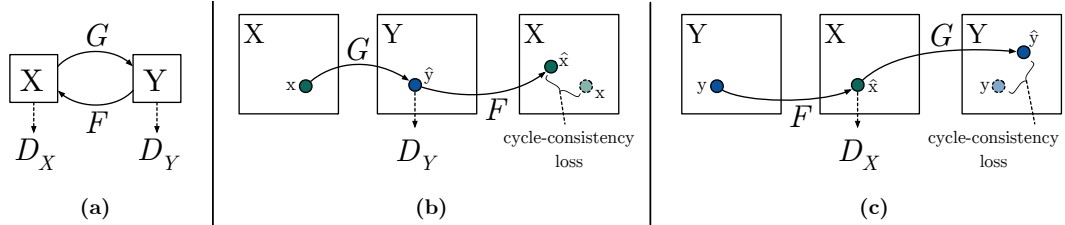

Figure 1: CycleGAN model.

such that a sample $x \in X$ is mapped to a sample $G(x) \in Y$ where

$$G(x) \in Y \text{ is the best representative of } x \text{ in } Y .\tag{1}$$

The mapping $G$ is typically approximated by a neural network $G_\theta$ parametrized by $\theta$. Without paired data, directly solving this is impossible but on a distribution level it is easily seen if $G$ solves eq. (1) then the distribution of $G(x)$ as $x$ is sampled from $X$ is equal to that of $Y$. Mathematically, if $X = (X, \mathcal{X}, \mu)$ and $Y = (Y, \mathcal{Y}, \nu)$ are probability spaces with probability measures $\mu$ and $\nu$ respectively, this can be written as

$$\nu(A) = \mu(\{x : G(x) \in A\}) = \mu(G^{-1}(A)) \stackrel{\text{def}}{=} (G_*\mu)(A) \quad \text{for all } A \in \mathcal{Y},\tag{2}$$

Or in words, the probability measure $\nu$ equals the push-forward measure $G_*\mu$. By Jensen's equality we can relate this to the fixed f-divergence $D_f$:

$$G_*\mu = \nu \text{ if and only if } D_f(G_*\mu \| \nu) = 0.\tag{3}$$

While adversarial adversarial optimization techniques such as GANs can in principle solve problem eq. (3), they remain under-constrained thus not giving a reasonable solution to the original problem eq. (1).

The idea behind the *cycle consistency condition* from (Zhu et al., 2017) is to enforce additional constraints by introducing another function $F : Y \to X$, which is also approximated by a neural network and tries to solve the inverse task: for each $y \in Y$ find $F(y) \in X$ that would be the best translation of $y$ to $X$. Similar to the reasoning above, this condition would imply that

$$\mu = F_*\nu \quad \text{and} \quad D_f(F_*\nu \| \mu) = 0.\tag{4}$$

The goal is to enforce that $F(G(x)) \approx x$ for all $x \in X$ and, similarly, that $G(F(y)) \approx y$ for all $y \in Y$, i.e. to minimize the following *cycle consistency loss*

$$\mathcal{L}_{\text{cyc}}(G, F) := \mathbb{E}_{x \sim X}\|(F \circ G)(x) - x\| + \mathbb{E}_{y \sim Y}\|(G \circ F)(y) - y\|,\tag{5}$$

where typically the $L^1$ norm is chosen, but in principle any norm can be chosen. Zhu et al. (Zhu et al., 2017) also suggested that an adversarial loss could in principle have been used here as well, but they did not note any performance improvement.

Combining these losses, we arrive at the *CycleGAN loss* defined as

$$\mathcal{L}(G, F) := D_f(F_*\mu \| \nu) + D_f(G_*\nu \| \mu) + \alpha_{\text{cyc}} \cdot \mathcal{L}_{\text{cyc}}(G, F),$$

where the factor $\alpha_{\text{cyc}} > 0$ determines the weight of the cycle consistency term. We illustrate the CycleGAN model in fig. 1.

Precautions with generative models have been addressed before, for example, unpaired image to image translation can hallucinate features in medical images (Cohen et al., 2018). Furthermore, it was already noted in (Zhu et al., 2017) that the CycleGAN might admit multpiple solutions and that the issue of tint shift in image-to-image translation arises due to the fact that for a fixed input image $x \in X$ multiple images $y_1, \ldots, y_n \in Y$ with different tints might be equally plausible. Adding identity loss term was suggested in (Zhu et al., 2017) to alleviate the tint shift issue, i.e., the *extended CycleGAN loss* is defined as

$$\mathcal{L}_{\text{ext}}(G, F) := \mathcal{L}(G, F) + \alpha_{\text{id}} \cdot (\mathbb{E}_{y \sim Y}\|F(y) - y\| + \mathbb{E}_{x \sim X}\|G(x) - x\|),$$

where the factor $\alpha_{\mathrm{id}} \geq 0$ determines the weight of the identity loss term. In general, to properly define the identity loss one needs to represent both $X$ and $Y$ as being the supported on the same manifold, which is limiting if the distributions are substantially different.

The goal of this work is to study the kernel, or null space, of the CycleGAN loss, which is the set of solutions $(G, F)$ which have zero 'pure' CycleGAN loss, and to give a perturbation bounds for approximate solutions for the case of extended CycleGAN loss. We do the theoretical analysis in section 2. We show that under certain assumptions on the probability spaces $X, Y$ the kernel has symmetries which allow for multiple possible solutions in Proposition 2.1. Furthermore, we show in Proposition 2.2 and the following remarks that the kernel admits a natural structure of a principle homogeneous space with the automorphism group $\mathrm{Aut}(X)$ of $X$ acting on the set of solutions freely and transitively. Next, we expand our analysis to the case of approximate solutions for the *extended* CycleGAN loss by proving perturbation bounds in Proposition 2.3 and Corollary 2.1. We discuss the existence problem of automorphism in Proposition 2.4 and Proposition 2.6. We proceed in section 3 by showing that unexpected symmetries can be *learned* by a CycleGAN. In particular, when translating the same domain to itself CycleGAN can learn a nontrivial automorphism of the domain. In appendix A, we briefly explain the measure-theoretic language we use heavily in the paper for those readers who are more used to working with distributions, and also remind the reader of some basic notions from differential geometry which we use as well.

## 2 THEORY

### 2.1 CYCLEGAN KERNEL AS A PRINCIPLE HOMOGENEOUS SPACE

The notions of isomorphism of probability spaces and of probability space automorphisms are central to this paper. Intuitively speaking, an isomorphism $f : X \to Y$ of probability spaces $X$ and $Y$ is a bijection between $X$ and $Y$ such that the probability of an event $A \subset Y$ equals the probability of event $\{x : F(\boldsymbol{x}) \in A\} \subset X$. An isomorphism of a probability space to itself is called a probability space automorphism. For example, if our probability space consists of samples from $n$-dimensional spherical Gaussian distribution, then any rotation in $\mathrm{SO}(\mathbb{R}^n)$ is a probability space automorphism. For a precise definition we refer the reader to appendix A.

Firstly, we prove that if at least one of the probability spaces $X, Y$ admits a nontrivial probability automorphism, then any exact solution in the kernel of CycleGAN can be altered giving a *different* solution.

**Proposition 2.1** (Invariance of the kernel). *Let* $X = (X, \mathcal{X}, \mu), Y = (Y, \mathcal{Y}, \nu)$ *be probability spaces and* $\varphi : X \to X$ *be a probability space automorphism. Let* $G : X \to Y$ *and* $F : Y \to X$ *be measurable maps satisfying*

$$\mathcal{L}(G, F) = 0. \tag{6}$$

*Then* $F, G$ *are probability space isomorphisms and*

$$\mathcal{L}(G \circ \varphi, \varphi^{-1} \circ F) = 0. \tag{7}$$

*If, furthermore,* $\varphi \neq \mathrm{id}_X$,[1] *then*

$$G \circ \varphi \neq G \quad and \quad \varphi^{-1} \circ F \neq F. \tag{8}$$

*Proof.* Since $\varphi$ is a probability space automorphism, its inverse $\varphi^{-1}$ is an automorphism as well. In particular, it is measure-preserving since

$$\mu(\varphi(A)) = \mu(\varphi^{-1}(\varphi(A))) = \mu(A) \quad \text{for all } A \in \mathcal{X}.$$

We note that by eq. (2) and the positivity of the norms eq. (6) implies that

$$G_*\mu = \nu, \quad F_*\nu = \mu \tag{9}$$

and

$$G \circ F = \mathrm{id}_Y \text{ a.e.}, \quad F \circ G = \mathrm{id}_X \text{ a.e.}. \tag{10}$$

---

[1] Inequality should be understood in the 'modulo null sets' sense here, i.e., we assert that there are positive probability sets on which the maps do differ.

Therefore both $F$ and $G$ are isomorphisms. By definition of $\mathcal{L}$,

$$\mathcal{L}(G \circ \varphi, \varphi^{-1} \circ F) = D_f((G \circ \varphi)_*\mu \| \nu) + D_f((\varphi^{-1} \circ F)_*\nu \| \mu)$$
$$+ \alpha_{\mathrm{cyc}} \cdot (\mathbb{E}_{\boldsymbol{x} \sim X} \| \varphi^{-1}(F(G(\varphi(\boldsymbol{x})))) - \boldsymbol{x}\| + \mathbb{E}_{\boldsymbol{y} \sim Y} \| G(\varphi(\varphi^{-1}(F(\boldsymbol{y})))) - \boldsymbol{y}\|).$$

Since $(G \circ \varphi)_*\mu = G_*(\varphi_*\mu)$ and $\varphi$ is measure-preserving, eq. (9) implies that $(G \circ \varphi)_*\mu = \nu$. Similarly, $(\varphi^{-1} \circ F)_*\nu = \mu$ since $\varphi^{-1}$ is measure-preserving as well. This shows that

$$D_f((G \circ \varphi)_*\mu \| \nu) = D_f((\varphi^{-1} \circ F)_*\nu \| \mu) = 0.$$

Using eq. (10) and the fact that $\varphi^{-1} \circ \varphi = \varphi \circ \varphi^{-1} = \mathrm{id}_X$ almost everywhere, we conclude that

$$\mathbb{E}_{\boldsymbol{y} \sim Y} \| G(\varphi(\varphi^{-1}(F(\boldsymbol{y})))) - \boldsymbol{y}\| = \mathbb{E}_{\boldsymbol{y} \sim Y} \| \boldsymbol{y} - \boldsymbol{y}\| = 0.$$

and

$$\mathbb{E}_{\boldsymbol{x} \sim X} \| \varphi^{-1}(F(G(\varphi(\boldsymbol{x})))) - \boldsymbol{x}\| = \mathbb{E}_{\boldsymbol{x} \sim X} \| \boldsymbol{x} - \boldsymbol{x}\| = 0.$$

Combining these observations together, we deduce that

$$\mathcal{L}(G \circ \varphi, \varphi^{-1} \circ F) = 0$$

and the proof of eq. (7) is complete. To prove eq. (8), first note that there exists a set $A \in \mathcal{X}$ such that $\mu(A) > 0$ and

$$\varphi(\boldsymbol{x}) \neq \boldsymbol{x} \quad \text{for all } \boldsymbol{x} \in A,$$

since we assume that $\varphi$ essentially differs from the identity mapping. If $G \circ \varphi = G$ $\mu$-a.e., then $F \circ G \circ \varphi = F \circ G$ $\mu$-a.e. as well, which implies that $\varphi(\boldsymbol{x}) = \boldsymbol{x}$ for $\mu$-almost every $\boldsymbol{x}$, which is a contradiction. In a similar way one can show that $\varphi^{-1} \circ F$ essentially differs from $F$. $\qquad\square$

We provide the following converse to Proposition 2.1.

**Proposition 2.2** (Kernel as a principle homogeneous space). *Let* $X = (X, \mathcal{X}, \mu), Y = (Y, \mathcal{Y}, \nu)$ *be probability spaces. Let* $F : X \to Y$, $G : Y \to X$ *and* $F' : X \to Y$, $G' : Y \to X$ *be measurable maps satisfying*

$$\mathcal{L}(F, G) = 0 \quad \text{and} \quad \mathcal{L}(F', G') = 0. \tag{11}$$

*Then there exists a unique probability space automorphism* $\varphi : X \to X$ *such that*

$$F \circ \varphi = F' \quad \text{and} \quad \varphi^{-1} \circ G = G'.$$

For the proof it suffices to take $\varphi := G \circ F'$. Combined with Proposition 2.1, this allows us to say that the group $\mathrm{Aut}(X)$ of probability space automorphisms of $X$ *acts freely and transitively* on the set of isomorphisms $\mathrm{Iso}(X, Y)$ when the latter set is nonempty. This amounts to saying that the space of solutions of CycleGAN is a principle homogeneous space. It can be helpful to view this result from the abstract category theory point of view, that is, if $\mathcal{C}$ is a category and $X \in \mathcal{C}$ is any fixed object, then for any object $Y \in \mathcal{C}$ the automorphism group $\mathrm{Aut}(X)$ acts on the set of homomorphisms $\mathrm{Hom}(X, Y)$ on the right by composition, i.e. we define

$$\alpha(\phi) := \phi \circ \alpha \quad \text{for all } \phi \in \mathrm{Hom}(X, Y), \ \alpha \in \mathrm{Aut}(X).$$

This action leaves the space of isomorphisms $\mathrm{Iso}(X, Y) \subseteq \mathrm{Hom}(X, Y)$ invariant, and this restricted action is transitive if $\mathrm{Iso}(X, Y)$ is nonempty, and, furthermore, free, i.e. $\alpha(\phi) \neq \phi$ for all $\alpha \neq \mathrm{id}_X$ and all $\phi \in \mathrm{Iso}(X, Y)$.

To proceed with our analysis for case of approximate solutions for extended CycleGAN loss, we first formulate a useful 'push-forward property' for general $f$-divergences between distributions on $\mathbb{R}^{n^2}$. The proof is provided in appendix A.

**Lemma 2.1** (Push-forward property for $f$-divergences). *Let* $p, q$ *be distributions on* $\mathbb{R}^n$ *and* $\varphi : \mathbb{R}^n \to \mathbb{R}^n$ *be a diffeomorphism. Then for any* $f$-divergence $D_f$ *we have*

$$D_f(\varphi_*p \| q) = D_f(p \| (\varphi^{-1})_*q) \tag{12}$$

We are now ready to prove the perturbation bounds for approximate solutions.

---

[2]While very natural to conjecture and easy to prove, we were unable to find references to it in existing ML literature, so we dubbed this property a 'push-forward property' and provide a proof.

**Proposition 2.3** (Perturbation bound). *Let* $X, Y$ *be probability spaces with probability densities* $p_X, p_Y \in L^1(\mathbb{R}^n)$ *and let* $\varphi \in \mathrm{Aut}(X)$ *be a diffeomorphic probability space automorphism. Assume that* $\varphi^{-1}$ *is* $C_\varphi$-*Lipshitz, where* $C_\varphi > 0$ *is some positive constant. Let* $G : \mathbb{R}^n \to \mathbb{R}^n$ *and* $F : \mathbb{R}^n \to \mathbb{R}^n$ *be measurable maps. Then the following perturbation bound holds for extended CycleGAN loss:*

$$\mathcal{L}_{ext}(G \circ \varphi, \varphi^{-1} \circ F) \le \max\left(C_\varphi, 1\right) \cdot \mathcal{L}_{ext}(G, F) + 2 \cdot \alpha_{id} \cdot \mathbb{E}_{\boldsymbol{x} \sim X} \|\varphi(\boldsymbol{x}) - \boldsymbol{x}\|. \tag{13}$$

*Proof.* The proof is an adaptation of the proof of Proposition 2.1. By definition of $\mathcal{L}_{ext}$,

$$\begin{aligned}
\mathcal{L}_{ext}(G \circ \varphi, \varphi^{-1} \circ F) =\ & D_f((G \circ \varphi)_* p_X \| p_Y) + D_f((\varphi^{-1} \circ F)_* p_Y \| p_X) \\
& + \alpha_{cyc} \cdot \left( \mathbb{E}_{\boldsymbol{x} \sim X} \|\varphi^{-1}(F(G(\varphi(\boldsymbol{x})))) - \boldsymbol{x}\| + \mathbb{E}_{\boldsymbol{y} \sim Y} \|G(\varphi(\varphi^{-1}(F(\boldsymbol{y})))) - \boldsymbol{y}\| \right) \\
& + \alpha_{id} \cdot \left( \mathbb{E}_{\boldsymbol{y} \sim Y} \|\varphi^{-1}(F(\boldsymbol{y})) - \boldsymbol{y}\| + \mathbb{E}_{\boldsymbol{x} \sim X} \|G(\varphi(\boldsymbol{x})) - \boldsymbol{x}\| \right).
\end{aligned}$$

Firstly, since $\varphi$ is measure-preserving, $D_f((G \circ \varphi)_* p_X \| p_Y) = D_f(G_* p_X \| p_Y)$. Using Lemma 2.1 and the fact that $\varphi$ is measure-preserving again, we see that

$$D_f((\varphi^{-1} \circ F)_* p_Y \| p_X) = D_f(F_* p_Y \| \varphi_* p_X) = D_f(F_* p_Y \| p_X).$$

Secondly,

$$\begin{aligned}
\mathbb{E}_{\boldsymbol{x} \sim X} \|\varphi^{-1}(F(G(\varphi(\boldsymbol{x})))) - \boldsymbol{x}\| &= \mathbb{E}_{\boldsymbol{x} \sim X} \|\varphi^{-1}(F(G(\varphi(\boldsymbol{x})))) - \varphi^{-1}(\varphi(\boldsymbol{x}))\| \\
&\overset{*}{=} \mathbb{E}_{\boldsymbol{x} \sim X} \|\varphi^{-1}(F(G(\boldsymbol{x}))) - \varphi^{-1}(\boldsymbol{x})\| \\
&\le C_\varphi \cdot \mathbb{E}_{\boldsymbol{x} \sim X} \|F(G(\boldsymbol{x})) - \boldsymbol{x}\|,
\end{aligned}$$

where the equality $(*)$ uses the fact that $\varphi$ is measure-preserving. As in before, $\mathbb{E}_{\boldsymbol{y} \sim Y} \|G(\varphi(\varphi^{-1}(F(\boldsymbol{y})))) - \boldsymbol{y}\| = \mathbb{E}_{\boldsymbol{y} \sim Y} \|G(F(\boldsymbol{y})) - \boldsymbol{y}\|$ since $\varphi \circ \varphi^{-1} = \mathrm{id}_X$ almost everywhere.

Finally, since $\varphi$ is a probability space automorphism and $\varphi^{-1}$ is $C_\varphi$-Lipshitz, we conclude that

$$\begin{aligned}
\mathbb{E}_{\boldsymbol{y} \sim Y} \|\varphi^{-1}(F(\boldsymbol{y})) - \boldsymbol{y}\| &\le \mathbb{E}_{\boldsymbol{y} \sim Y} \|\varphi^{-1}(F(\boldsymbol{y})) - \varphi^{-1}(\boldsymbol{y})\| + \mathbb{E}_{\boldsymbol{y} \sim Y} \|\varphi^{-1}(\boldsymbol{y}) - \boldsymbol{y}\| \\
&\le C_\varphi \cdot \mathbb{E}_{\boldsymbol{y} \sim Y} \|F(\boldsymbol{y}) - \boldsymbol{y}\| + \mathbb{E}_{\boldsymbol{y} \sim Y} \|\varphi^{-1}(\boldsymbol{y}) - \boldsymbol{y}\| \\
&= C_\varphi \cdot \mathbb{E}_{\boldsymbol{y} \sim Y} \|F(\boldsymbol{y}) - \boldsymbol{y}\| + \mathbb{E}_{\boldsymbol{y} \sim Y} \|\varphi(\boldsymbol{y}) - \boldsymbol{y}\|
\end{aligned}$$

and that

$$\begin{aligned}
\mathbb{E}_{\boldsymbol{x} \sim X} \|G(\varphi(\boldsymbol{x})) - \boldsymbol{x}\| &= \mathbb{E}_{\boldsymbol{x} \sim X} \|G(\varphi(\boldsymbol{x})) - \varphi^{-1}(\varphi(\boldsymbol{x}))\| = \mathbb{E}_{\boldsymbol{x} \sim X} \|G(\boldsymbol{x}) - \varphi^{-1}(\boldsymbol{x})\| \\
&\le \mathbb{E}_{\boldsymbol{x} \sim X} \|G(\boldsymbol{x}) - \boldsymbol{x}\| + \mathbb{E}_{\boldsymbol{x} \sim X} \|\boldsymbol{x} - \varphi^{-1}(\boldsymbol{x})\| \\
&= \mathbb{E}_{\boldsymbol{x} \sim X} \|G(\boldsymbol{x}) - \boldsymbol{x}\| + \mathbb{E}_{\boldsymbol{x} \sim X} \|\varphi(\boldsymbol{x}) - \boldsymbol{x}\|.
\end{aligned}$$

Combining all these estimates together, we deduce that

$$\mathcal{L}_{ext}(G \circ \varphi, \varphi^{-1} \circ F) \le \max\left(C_\varphi, 1\right) \cdot \mathcal{L}_{ext}(G, F) + 2 \cdot \alpha_{id} \cdot \mathbb{E}_{\boldsymbol{x} \sim X} \|\varphi(\boldsymbol{x}) - \boldsymbol{x}\|$$

and the proof is complete. $\qquad\square$

**Corollary 2.1** (Asymptotic perturbation bound). *In the setting of Proposition 2.3, let* $G_i : \mathbb{R}^n \to \mathbb{R}^n$ *and* $F_i : \mathbb{R}^n \to \mathbb{R}^n$ *for* $i \ge 1$ *be a sequence of measurable maps such that the 'pure' CycleGAN loss converges to zero, i.e.,*

$$\lim_{i \to \infty} \mathcal{L}(G_i, F_i) = 0$$

*and let*

$$\overline{\mathcal{L}}_{id} := \limsup_{i \to \infty} \left( \mathbb{E}_{\boldsymbol{y} \sim Y} \|F_i(\boldsymbol{y}) - \boldsymbol{y}\| + \mathbb{E}_{\boldsymbol{x} \sim X} \|G_i(\boldsymbol{x}) - \boldsymbol{x}\| \right).$$

*Then the following asymptotic perturbation bound holds for the 'extended' CycleGAN loss:*

$$\limsup_{i \to \infty} \mathcal{L}_{ext}(G_i \circ \varphi, \varphi^{-1} \circ F_i) \le \max\left(C_\varphi, 1\right) \cdot \alpha_{id} \cdot \overline{\mathcal{L}}_{id} + 2 \cdot \alpha_{id} \cdot \mathbb{E}_{\boldsymbol{x} \sim X} \|\varphi(\boldsymbol{x}) - \boldsymbol{x}\|.$$

Corollary 2.1 has a direct practical implication. When using a CycleGAN model for translating substantially different distributions (such as different medical imaging modalities) one would be forced to pick a small value for $\alpha_{id}$ in order for the model to produce reasonable results. Furthermore, since the distributions are substantially different, we can expect that $\overline{\mathcal{L}}_{id} \gg 2 \cdot \mathbb{E}_{\boldsymbol{x} \sim X} \|\varphi(\boldsymbol{x}) - \boldsymbol{x}\|$ for many nontrivial automorphism $\varphi$. Therefore, the asymptotic perturbation bound automatically implies that the approximate solution space admits a lot of symmetry, potentially leading to undesirable results.

## 2.2 EXISTENCE OF AUTOMORPHISMS

By Proposition 2.1 we see that if either space admits a nontrivial probability automorphism, then the CycleGAN problem has multiple solutions. However, for this to be a problem in practice there must actually exist such probability automorphisms, which we shall now show is the case. First of all, we state the following proposition, which says that we can transfer automorphism from an isomorphic copy of X to X itself.

**Lemma 2.2.** *Let $f : Z \to X$ be an isomorphism of probability spaces and $T : Z \to Z$ be an automorphism of Z. Then $S := f \circ T \circ f^{-1}$ is an automorphism of X and the diagram*

$$
\begin{array}{ccc}
Z & \xleftarrow{\ f^{-1}\ } & X \\
{\scriptstyle T}\downarrow & & \downarrow{\scriptstyle S} \\
Z & \xrightarrow{\ f\ } & X
\end{array}
$$

*commutes. Furthermore, if $Z \subset \mathbb{R}^n$, $X \subset \mathbb{R}^m$ are submanifolds and $f$, $T$ are diffeomorphisms, then $S$ is a diffeomorphism as well.*

*Proof.* The first claim follows from invertibility of $f$ and $T$. The second claim follows from the definition of a diffeomorphism between submanifolds, see appendix A. $\qquad\square$

An important notion in probability theory is that of a Lebesgue probability space. Many probability spaces which emerge in practice such as $[0, 1]^n \subset \mathbb{R}^n$ with the Lebesgue measure or $\mathbb{R}^n$ with a Gaussian probability distribution, both defined on the respective $\sigma$-algebras of Lebesgue measurable sets, are instances of Lebesgue probability spaces.

**Definition 2.1.** *A probability space X is called a Lebesgue probability space if it is isomorphic as a measure space to a disjoint union $([0, c], \lambda)$, where $\lambda$ is the Lebesgue measure on the $\sigma$-algebra of Lebesgue measurable subsets of the interval $[0, c]$, and at most countably many atoms of total mass $1 - c$.*

Informally speaking, this definition says that Lebesgue probability spaces consist of a continuous part and at most countably many Dirac deltas (=atoms). First of all, we provide an abstract result about existence of nontrivial probability space automorphisms in Lebesgue probability spaces which are either 'not purely atomic' or have at least two atoms with equal mass. 'Not purely atomic' means that the sum of the probabilities of all atoms is strictly less than 1.

**Proposition 2.4.** *Let X be a Lebesgue probability space such that at least one of the assumptions*

1. *X not purely atomic;*

2. *there exist at least two atoms $a_j, a_k$ in X with equal mass*

*holds. Then X admits nontrivial automorphisms.*

*Proof.* If the space X is not purely atomic, we have $X \simeq [0, c] \sqcup \bigsqcup_{i \geq 1} a_i$ for some $c > 0$, where $[0, c]$ is the continuous part and $\bigsqcup_{i \geq 1} a_i$ is the atomic part of the probability measure $\mu$. Interval $[0, c]$ admits at least one nontrivial automorphism, namely the transformation $x \mapsto c - x$ (leaving the atoms fixed), hence so does X by Lemma 2.2. In fact, there are infinitely many other automorphisms, which can be obtained by exchanging nonoverlapping subintervals $(a, a + d), (b, b + d) \subset [0, c]$ of the same length. If there exist two atoms $a_j, a_k$ in X with equal mass, then a transformation which transposes $a_j$ with $a_k$ and keeps the rest of X fixed is a nontrivial automorphism. $\qquad\square$

Probability spaces of images which appear in real life typically have a continuous component which would correspond to continuous variations in object sizes, lighting conditions, etc. Therefore, they admit some probability space automorphisms. However, such abstract automorphisms can be highly discontinuous, which would make it questionable if neural networks can learn them. We would like to show that there are also automorphisms which are smooth, at least locally. For this, we first state the following technical claim. The proof is provided in appendix A.

**Proposition 2.5.** *Let $\mu$ be a Borel probability measure on $\mathbb{R}^n$ and $f : \mathbb{R}^n \to \mathbb{R}^m$ be a continuous injective function. Then $f : (\mathbb{R}^n, \mathcal{B}(\mathbb{R}^n), \mu) \to (\mathbb{R}^m, \mathcal{B}(\mathbb{R}^m), f_*\mu)$ is an isomorphism of probability spaces, where $f_*\mu$ denotes the push-forward of measure $\mu$ to $\mathbb{R}^m$.*

Finally, we show the existence of smooth automorphisms under the assumption that our data manifold $\mathcal{D} \subset \mathbb{R}^m$ can be generated by embedding $\mathbb{R}^n$ with standard Gaussian measure into $\mathbb{R}^m$ as a submanifold. We write $\gamma_n$ for the standard Gaussian probability measure on the space $\mathbb{R}^n$.

**Proposition 2.6.** *Let $\mathrm{Z} := (\mathbb{R}^n, \mathcal{B}(\mathbb{R}^n), \gamma_n)$ be an $n$-dimensional standard Gaussian distribution. Let $f : \mathbb{R}^n \to \mathbb{R}^m$ be a manifold embedding. Denote by $\mathrm{X}$ the probability space $(\mathbb{R}^m, \mathcal{B}(\mathbb{R}^m), f_*\gamma_n)$. Then the following assertions hold:*

1. *$f$ is an isomorphism of probability spaces when viewed as a map $\mathrm{Z} \to \mathrm{X}$;*

2. *every rotation $T \in \mathrm{SO}(\mathbb{R}^n)$ is a probability space automorphism and a diffeomorphism of $\mathrm{Z}$. $T$ induces a probability space automorphism of $\mathrm{X}$ which is, additionally, a diffeomorphism when restricted to $\mathrm{Im}\, f \subset \mathbb{R}^m$.*

*Proof.* The first claim follows directly from Proposition 2.5. For the second part, it is clear that rotations in $\mathrm{SO}(\mathbb{R}^n)$ preserve isotropic Gaussian distribution, and the rest follows from Lemma 2.2. $\square$

The connection with generative models is clear if we take $f$ to be an invertible generative model such as RealNVP Dinh et al. (2016) or Glow Kingma & Dhariwal (2018). The assumption of manifold embedding in the proposition can be seen as too limiting in general, and we explain how to 'bypass' it in Lemma A.2 for the interested readers. In conclusion, if we assume that the distributions we are working with could be represented by an invertible generative model, then there exists a rich space of automorphisms. Given the success of e.g. Glow, this assumption seems to be valid for natural images.

## 3 NUMERICAL RESULTS

Since we have established that the existence of automorphisms can negatively impact the results of CycleGAN, we now demonstrate how this can happen by considering a toy case with a known solution and demonstrating that CycleGAN can and does learn a nontrivial automorphism. The toy experiment which we perform is translation of MNIST dataset to itself. That is, at training time we pick two minibatches batch$_A$ and batch$_B$ from MNIST at random and use these as samples from $\mathrm{X}$ and $\mathrm{Y}$ respectively. The generator neural network in this case is a convolutional autoencoder with residual blocks, fully connected layer in the bottleneck and *no* skip connections from encoder to decoder. We also train a simple CNN for MNIST classification in order to classify CycleGAN outputs. The networks were trained using SGD. The 'natural' transformation in this case is, of course, the identity mapping and we expect the classification of the inputs and outputs to stay the same. But we shall see that this is not the case.

In fig. 2a–fig. 2h we show some examples for the generated fake samples and the reconstruction on test set. In fig. 3a–fig. 3b we provide the confusion matrices for the A2B and B2A generators respectively. We use these matrices to understand if e.g. the class of transformed image for A2B translation equals the source class, or if is a random variable independent of the source class, or if we can spot some deterministic permutation of classes. We have observed that in practice the identity mapping is not learned. Instead, the network leans towards producing a certain permutation of digits, rather than identity or a random assignment of classes independent of the source label. One explanation would be as follows. Suppose that we can perfectly disentangle class and style in latent digit representation Makhzani et al. (2015). Then *any* permutation in $S_{10}$, acting on the class part of the latent code, determines a probability space automorphism on the space of digits, which can be learned by a neural network. Further investigation of confusion matrices reveals that the networks introduce short *cycles*, e.g., mapping 2 to 6 and vice versa.

We provide additional experiments on BRATS2015 dataset in appendix B, where we show that in the absense of identity loss the pure CycleGAN loss demonstrates noticeable symmetry, while the PSNR is clearly not invariant. Increasing the weight of the identity loss term reduces the symmetry, but does not necessarily result in a similar PSNR improvement.

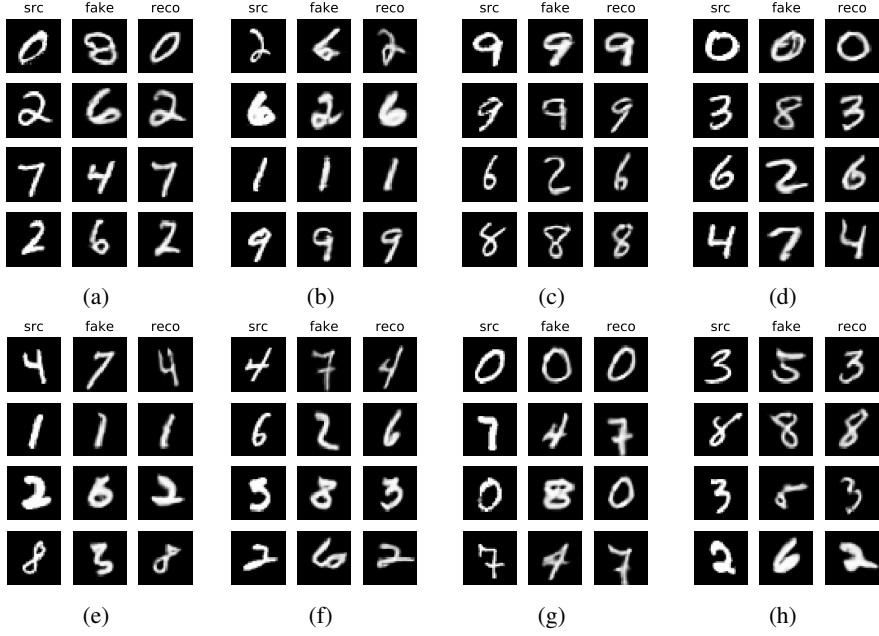

Figure 2: Examples on MNIST2MNIST task. (a)-(d) A2A translation, first column are samples from A, second column are 'fake B' and third column are reconstructions of original samples from A (e)-(h) same for B2B translation.

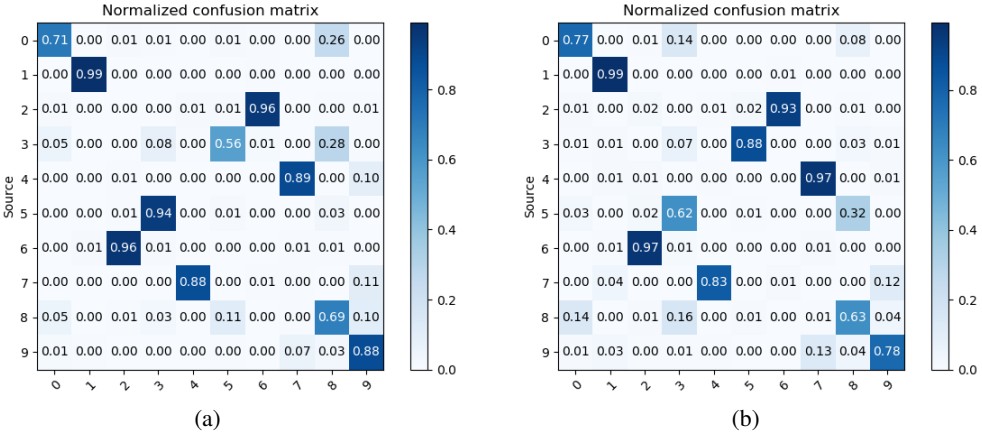

Figure 3: Normalized confusion matrices for A2B and B2A generator respectively.

## 4 DISCUSSION AND FUTURE WORK

We have shown theoretically that under mild assumptions, the kernel of the CycleGAN admits nontrivial symmetries and has a natural structure of a principle homogeneous space. To show empirically that such symmetries can be learned, we have trained a CycleGAN on the task of translating a domain to itself. In particular, we show that on the MNIST2MNIST task, in contrast to the expected identity, the CycleGAN learns to permute the digits. We have therefore effectively shown, that it is not the CycleGAN loss which prevents this from occurring more often, but hypothesize that the network architecture also has major influence. We advocate against the usage of CycleGAN when translating between substantially different distributions in critical tasks such as medical imaging, given the theoretical results in Corollary 2.1 which suggest ambiguity of solutions, even in the presence of the identity loss term.

We would like to point out that some work has been done recently extending the CycleGAN. For example, in Na et al. (2019) the authors argue that many image-to-image translation tasks are 'multimodal' in a sense that there are multiple equally plausible outputs for a single input image, therefore, one should explicitly model this uncertainty in the model. To address this issue, the authors design a network which has two 'style' encoders $E_X : X \to Z_X, E_Y : Y \to Z_Y$, two discriminators for each domain, two conditional encoders for each direction $E_{XY} : X \times Z_Y \to Z_{XY}, E_{YX} : Y \times Z_X \to Z_{YX}$ and two generators for each direction $G_{XY} : Z_{XY} \to Y, G_{YX} : Z_{YX} \to X$. The style encoders serve to extract the 'style' of the image, which is present in both domains, e.g., in case of the 'female-to-male' task on CelebA dataset the style would correspond to coarsely represented facial features. The loss term forces the mutual information between the style vector of the translated image and the input style to the conditional encoder to be maximized. This allows the network to roughly preserve the style in the translation. While we leave full analysis of this approach for the future work, we expect that such loss would reduce ambiguity in the solution space to those isomorphisms which differ by automorhpishs from the set

$$\{\varphi \in \mathrm{Aut}(X) : E_X \circ \varphi(x) = E_X(x)\}$$

leaving the style fixed, since replacing $G_{YX}$ with $\varphi \circ G_{YX}$ and $E_{XY}$ with $E_{XY} \circ \varphi^{-1}$ does not change the loss value for such $\varphi$. Therefore, the reduction in uncertainty of our solution depends on capacity of the encoder $E_X$, and, ideally, should be quantified. In particular, one might still need to enforce additional problem-specific features in the encoder $E_X$ to guarantee that important image style content is preserved.

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

## A    BACKGROUND

Firstly, we very briefly explain the probability theory language we use in this article, and we refer the reader to (Eisner et al., 2015; Bogachev, 2007) for more details. Formally, a *measurable space* $(X, \mathcal{X})$ is a pair of a set $X$ and a $\sigma$-algebra $\mathcal{X}$ of subsets of $X$. Given a topological space $X$ with topology $\mathcal{U}$, there exists the smallest $\sigma$-algebra $\mathcal{B}(X)$, which contains all open sets in $\mathcal{U}$. This $\sigma$-algebra is called *Borel $\sigma$-algebra* of $X$ and its elements are called *Borel* sets. A *probability space* $\mathrm{X} = (X, \mathcal{X}, \mu)$ is a triple of a set $X$, a sigma algebra $\mathcal{X}$ of subsets of $X$ and a probability measure $\mu$ defined on the sigma-algebra $\mathcal{X}$. Given a probability space $(X, \mathcal{X}, \mu)$, a measurable set $A \in \mathcal{X}$ is called an *atom* if $\mu(A) > 0$ and for all measurable $B \subset A$ such that $\mu(B) < \mu(A)$ we have $\mu(B) = 0$. Given measurable spaces $(X, \mathcal{X})$ and $(Y, \mathcal{Y})$, we say that a mapping $\phi : X \to Y$ is *measurable* if for any $A \in \mathcal{Y}$ we have $\phi^{-1}(A) \in \mathcal{X}$. If $\mathrm{X} = (X, \mathcal{X}, \mu)$ and $\mathrm{Y} = (Y, \mathcal{Y}, \nu)$ are probability spaces and $\phi : X \to Y$ is a measurable map, we say that $\phi$ is *measure-preserving* if for all $A \in \mathcal{Y}$ we have $\mu(\phi^{-1}(A)) = \nu(A)$. An approximation argument easily shows that a measurable transformation $\phi : X \to X$ is measure-preserving if and only if for all nonnegative measurable functions $f$ on $X$ we have

$$\mathbb{E}_{\boldsymbol{x} \sim \mathrm{X}} f(\boldsymbol{x}) d\mu = \mathbb{E}_{\boldsymbol{x} \sim \mathrm{X}} (f \circ \phi)(\boldsymbol{x}) d\mu.$$

Given a probability space $\mathrm{X}$, a measurable space $(Y, \mathcal{Y})$ and a measurable map $\phi : X \to Y$, we define the *push-forward measure* $\phi_* \mu$ on $\mathcal{Y}$ by setting $(\phi_* \mu)(A) := \mu(\phi^{-1}(A))$ for all $A \in \mathcal{Y}$.

Let $(X, \mathcal{X}, \mu)$ and $(Y, \mathcal{Y}, \nu)$ be probability spaces and $f : X \to Y$ be a measure-preserving map. A measurable map $g : Y \to X$ is called an *essential inverse* of $f$ if $f \circ g = \mathrm{id}_Y$ for $\nu$-almost every $\boldsymbol{y} \in Y$ and $g \circ f = \mathrm{id}_X$ for $\mu$-almost every $\boldsymbol{x} \in X$. One can show that essential inverse is measure preserving and uniquely defined up to equality almost everywhere. We say that $f$ is an *isomorphism* if it admits an essential inverse. An isomorphism $f : X \to X$ is called an *automorphism*.

**Lemma A.1** (Push-forward property for $f$-divergences)**.** *Let $p, q$ be distributions on $\mathbb{R}^n$ and $\varphi : \mathbb{R}^n \to \mathbb{R}^n$ be a diffeomorphism. Then for any $f$-divergence $D_f$ we have*

$$D_f(\varphi_* p \| q) = D_f(p \| (\varphi^{-1})_* q) \tag{14}$$

*Proof.* First of all, change of variables formula for the integral implies that

$$(\varphi_* p)(\boldsymbol{x}) = p(\varphi^{-1}(\boldsymbol{x})) \left| \det \frac{\partial \varphi^{-1}}{\partial \boldsymbol{x}} \right| (\boldsymbol{x}) \quad \text{for all } \boldsymbol{x} \in \mathbb{R}^n,$$

$$((\varphi^{-1})_* q)(\boldsymbol{y}) = q(\varphi(\boldsymbol{y})) \left| \det \frac{\partial \varphi}{\partial \boldsymbol{y}} \right| (\boldsymbol{y}) \quad \text{for all } \boldsymbol{y} \in \mathbb{R}^n.$$

Therefore,

$$D_f(\varphi_* p \| q) = \int f \left( \frac{\varphi_* p(\boldsymbol{x})}{q(\boldsymbol{x})} \right) q(\boldsymbol{x}) d\boldsymbol{x} = \int f \left( \frac{p(\varphi^{-1}(\boldsymbol{x})) \left| \det \partial \varphi^{-1} / \partial \boldsymbol{x} \right| (\boldsymbol{x})}{q(\boldsymbol{x})} \right) q(\boldsymbol{x}) d\boldsymbol{x}.$$

Applying change of variables formula with $\boldsymbol{x} = \varphi(\boldsymbol{y})$, we get

$$\int f \left( \frac{p(\varphi^{-1}(\boldsymbol{x})) \left| \det \partial \varphi^{-1} / \partial \boldsymbol{x} \right| (\boldsymbol{x})}{q(\boldsymbol{x})} \right) q(\boldsymbol{x}) d\boldsymbol{x}$$

$$= \int f \left( \frac{p(\varphi^{-1}(\varphi(\boldsymbol{y}))) \left| \det \partial \varphi^{-1} / \partial \boldsymbol{x} \right| (\varphi(\boldsymbol{y}))}{q(\varphi(\boldsymbol{y}))} \right) q(\varphi(\boldsymbol{y})) \left| \det \frac{\partial \varphi}{\partial \boldsymbol{y}} \right| (\boldsymbol{y}) d\boldsymbol{x}$$

$$\overset{*}{=} \int f \left( \frac{p(\boldsymbol{y})}{q(\varphi(\boldsymbol{y})) \left| \det \partial \varphi / \partial \boldsymbol{y} \right| (\boldsymbol{y})} \right) q(\varphi(\boldsymbol{y})) \left| \det \frac{\partial \varphi}{\partial \boldsymbol{y}} \right| (\boldsymbol{y}) d\boldsymbol{x} = D_f(p \| (\varphi^{-1})_* q),$$

where the equality in $(*)$ uses a general property of Jacobians of smooth invertible maps that $\frac{\partial \varphi^{-1}}{\partial \boldsymbol{x}} \circ \varphi = \left( \frac{\partial \varphi}{\partial \boldsymbol{y}} \right)^{-1}$. Hence $D_f(\varphi_* p \| q) = D_f(p \| (\varphi^{-1})_* q)$, which completes the proof.    $\square$

We remind the reader that a *Polish space* is a separable completely metrizable topological space. A *Borel probability space* is a Polish space endowed with a probability measure $\mu$ on its Borel $\sigma$-algebra,

and we will also say that $\mu$ is a *Borel probability measure*. The basic examples of Borel probability spaces would be e.g. the spaces $[0,1]^n \subset \mathbb{R}^n$ with its Borel $\sigma$-algebra $\mathcal{B}(\mathbb{R}^n)$, endowed with Lebesgue measure $\lambda_n$. A Borel $\sigma$-algebra of the space $[0,1]^n$ endowed with Lebesgue measure $\lambda_n$ can be extended by adding all $\lambda_n$-measurable sets, leading to the $\sigma$-algebra of *Lebesgue-measurable sets*.

For the proof of Proposition 2.5 we need the following theorem, see Kechris (1995), Theorem 15.1.

**Theorem A.1** (Lusin-Souslin theorem). *Let $X, Y$ be Polish spaces and $f : X \to Y$ be continuous. If $A \subset X$ is Borel and $f|_A$ is injective, then $f(A)$ is Borel.*

*Proof of Proposition 2.5.* Denote the image $f(\mathbb{R}^n) \subset \mathbb{R}^m$ by $\mathrm{Im}\, f$. Then $\mathrm{Im}\, f \subset \mathbb{R}^m$ is a Borel subset, since $\mathbb{R}^n$ is a countable union of a compact sets and $f$ is continuous. Furthermore, from Lusin-Souslin theorem (theorem A.1) it follows that for every Borel subset $A \subset \mathbb{R}^n$ its image $f(A) \subset \mathbb{R}^m$ is Borel as well. Pick a point $x_0 \in \mathbb{R}^n$ which is not an atom of $\mu$. We want to define an almost everywhere inverse $\tilde{f}$ of $f$. Define a function $\tilde{f} : \mathbb{R}^m \to \mathbb{R}^n$ by

$$\tilde{f}(\boldsymbol{x}) = \begin{cases} f^{-1}(\boldsymbol{x}), & \text{if } \boldsymbol{x} \in \mathrm{Im}\, f. \\ x_0, & \text{otherwise.} \end{cases}$$

Using the remark above it is easy to see that $\tilde{f}$ is Borel measurable and that $(f_*\mu)(\tilde{f}^{-1}(A)) = \mu(A)$ for every Borel $A$. It follows from the definition that $\tilde{f} \circ f = \mathrm{id}_{\mathbb{R}^n}$ and that

$$f \circ \tilde{f}(\boldsymbol{x}) = \begin{cases} x, & \text{if } \boldsymbol{x} \in \mathrm{Im}\, f. \\ f(x_0), & \text{otherwise.} \end{cases}$$

Since $(f_*\mu)(\mathrm{Im}\, f) = 1$, $\tilde{f}$ is an almost everywhere inverse to $f$. We conclude that $f$ is a probability space isomorphism. $\square$

Secondly, we remind the reader of a couple of notions from differential geometry which we use in the text, and we refer the reader to e.g. (Warner, 1983) for more details. Given a subset $X$ of a manifold $M$ and a subset $Y$ of a manifold $N$, a function $f : X \to Y$ is said to be smooth if for all $p \in X$ there is a neighborhood $U \subset M$ of $p$ and a smooth function $g : U \to N$ such that $g$ extends $f$, i.e., the restrictions agree $g|_{U \cap X} = f|_{U \cap X}$. $f$ is said to be a *diffeomorphism* between $X$ and $Y$ if it is bijective, smooth and its inverse is smooth. Let $M$ and $N$ be smooth manifolds. A differentiable mapping $f : M \to N$ is said to be an *immersion* if the tangent map $d_p f : T_p M \to T_{f(p)} N$ is injective for all $p \in M$. If, in addition, $f$ is a homeomorphism onto $f(M) \subset N$, where $f(M)$ carries the subspace topology induced from $N$, we say that $f$ is an *embedding*. If $M \subset N$ and the inclusion map $\imath : M \to N$ is an embedding, we say that $M$ is a *submanifold* of $N$. Thus, the domain of an embedding is diffeomorphic to its image, and the image of an embedding is a submanifold.

We close this section with a small lemma, explaining how one can weaken the embedding assumption for generative models in Proposition 2.6.

**Lemma A.2.** *Let $f : \mathbb{R}^n \to \mathbb{R}^m$ be an injective manifold immersion. Let $B_R \subset \mathbb{R}^n$ be an open ball of radius $R > 0$ in $\mathbb{R}^n$ and $\overline{B}_R$ be its closure. Then $f : B_R \to f(B_R)$ is a manifold embedding.*

*Proof.* Since $\overline{B}_R$ is compact and $f$ is continuous, image of every closed subset $A \subseteq \overline{B}_R$ is compact and hence closed. This shows that $f^{-1} : f(\overline{B}_R) \to \overline{B}_R$ is continuous and thus $f : \overline{B}_R \to f(\overline{B}_R)$ is a homeomorphism. Restricting to the open ball $B_R \subset \overline{B}_R$, we conclude that $f : B_R \to f(B_R)$ is a homemorphism and thus a manifold embedding. $\square$

As a consequence, for our example with spherical Gaussian latent vector one can take sufficiently large ball of radius $R > 0$ in the latent space, truncating the latent distribution to 'sufficiently likely' values. This ball remains invariant under rotations, thus leading to a differentiable automorphism on the submanifold of 'sufficiently likely' images.

# B  BRATS2015 EXPERIMENTS

We present some additional results on the BRATS2015 dataset. For this experiment Unet-based generators with residual connections were used. The number of downsampling layers was 4 for

both generators, and skip connections were preserved. We trained all models for 20 epochs with Adam optimizer and learning rate 0.0002. We trained 4 models with $\alpha_{id} \in \{0.0, 10.0, 20.0, 40.0\}$. No data augmentation was used so as to avoid creating any additional symmetries. All images were normalized by dividing by the $95\%$-percentile, as is common in medical imaging when working with MR data.

We hypothesize that flipping images horizontally is a distribution symmetry. We measure the final test loss for both the network output (**Loss**) and its flipped version (**Loss (f)**), as well as the PSNR for both translation directions without (**PSNR T1-Fl**, **PSNR Fl-T1**) and with horizontal flips (**PSNR T1-Fl (f)**, **PSNR Fl-T1 (f)**). We summarize these results in table 1.

We observe that in the absense of identity loss the pure CycleGAN loss demonstrates noticeable symmetry, while the PSNR is clearly not invariant. Increasing the weight of the identity loss term reduces the symmetry, but does not always result in a similar PSNR improvement. We present some samples from the model with $\alpha_{id} = 0$ in fig. 4a, fig. 4b.

Table 1: Results on BRATS2015

| $\alpha_{id}$ | Loss | Loss (f) | PSNR T1-Fl | PSNR T1-Fl (f) | PSNR Fl-T1 | PSNR Fl-T1 (f) |
|---|---|---|---|---|---|---|
| 0.0 | 0.83 | 1.01 | 23.8 | 15.4 | 26.2 | 15.6 |
| 10.0 | 0.46 | 2.31 | 24.6 | 15.5 | 27.1 | 16.0 |
| 20.0 | 0.93 | 4.62 | 24.0 | 15.2 | 26.7 | 15.8 |
| 40.0 | 3.36 | 11.27 | 24.6 | 16.0 | 27.0 | 16.0 |

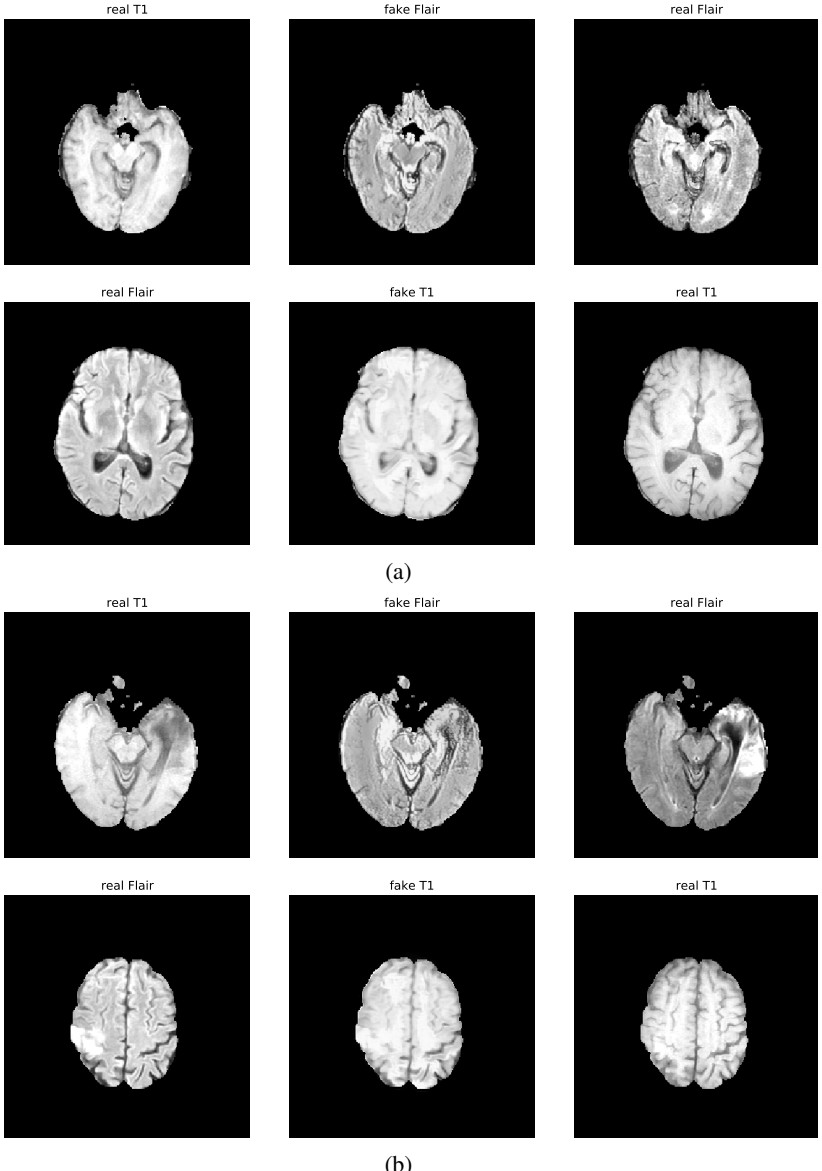

Figure 4: T1-Flair and Flair-T1 translation samples.

