# OpenReview forum: "Kernel of CycleGAN as a principal homogeneous space"
_ICLR.cc/2020/Conference — Accept (Poster)_

### Official Review · AnonReviewer2 · 2019-10-23
**Official Blind Review #2**

**Rating:** 3

**Review:**

I have read the rebuttal of the authors . Thank you for you answer and for addressing some concerns.  While the question addressed is important, the theory presented here does not seem to hint to a solution, hence I am keeping my score.

###
Summary of the paper:

This paper shows that the cycle GAN loss suffers from the presence of lot of symmetries that make the existence of a unique solution not possible , and moreover adding a regularizer that uses the identity loss is not enough to make the problem less prone to those invariances.

Review of the paper:

The notations and the formalism  in the paper are heavy and cumbersome and don't come with any surprising result, since the transforms between unpaired spaces will be found always up to   symmetries since we have the composition of one map with another. The use of the identity loss is also shown to not to help either in fixing this invariance issue.

Experiments are not interesting since without any structure on the map of F and G , the source domain and the target domain, one is expected to get permutations.

The paper points in the conclusion  that the use of skipconnection in F and G has the major influence.

 The study of cycle GAN might need some assessment of what is the mutual information between the domains , as on what  information needs be preserved , and information needs to match , skip connection maintain the content in image generation as the information is kept from lower layer and its modified to target the style of the target images.

An information theoretic analysis of cycle gan is needed using for example the objective of "MISO: Mutual Information Loss with Stochastic Style Representations for Multimodal Image-to-Image Translation".
or by using a radically new approach  for cycle gan such as the Gromov Wasserstein distance as done in " Learning Generative Models Across Incomparable Spaces"


**Experience Assessment:**

I have published one or two papers in this area.

**Review Assessment: Checking Correctness Of Derivations And Theory:**

I carefully checked the derivations and theory.

**Review Assessment: Checking Correctness Of Experiments:**

I carefully checked the experiments.

**Review Assessment: Thoroughness In Paper Reading:**

I read the paper thoroughly.

---

> ### Author Response · Authors · 2019-11-15
> **Thank you for your feedback!**
>
> Thank you very much for your feedback and additional references, this is very interesting!
>
> We opted to use the measure theory language because the language of probability distributions is not flexible enough to accomodate commonly occurring distributions, e.g. those (strictly) supported on lower dimensional manifolds. Unless we require the PDF explicitly to have some symmetries, it is not clear why the corresponding probability space would have any smooth automorphisms at all. Thus we show that the existence of automorphisms is a very general property, and in the setup of e.g. latent space with spherical Gaussian PDF we show that there are smooth automorphisms as well.
>
> We would like to point out that the goal of the paper is to provide a well-grounded and mostly self-contained analysis for the basic CycleGAN approach and to analyze theoretically the effect of the commonly used identity loss, along with some experiments to justify the claims. While the problem of multiple solutions for the CycleGAN is commonly realised, a good theoretical explanation for this is lacking in ML literature.
>
> Naturally, there are other approaches to unsupervised image-to-image translation with different losses and architectures. While analyzing all of them in a single paper is not realistic, we think that the philosophy we suggest in this paper can help researchers better understand the potential and the limitations of these newer image-to-image translation models. The underlying automorphisms can always pose a problem, and the question then becomes if a new loss/new architecture explicitly restricts this set.
>
> We have added some additional experiments on BRATS 2015 dataset. In this set of experiments we will show how the loss and PSNR (since we have a ground truth) change when we vary the weight for identity loss. We introduce a (approximate) probability automorphism in the form of left/right flips and show that this highly unwanted transformation still obtains low loss values unless an identity loss is used.
>
> We have also added a discussion about the MISO paper you suggested, where we hypothesize that the MISO approach does not 'solve' the issue of unwanted automorphisms, but rather restricts the set of these automorphisms to those that leave the style of the image fixed. Therefore the amount of uncertainty in this solution is connected to the capacity of the style encoder, and should ideally be quantified. When some important style content is present - e.g., anatomical landmarks - it seems reasonable that one should make sure that the style encoder learns this information. We think it is a an interesting question for future work.

---

### Official Review · AnonReviewer1 · 2019-10-23
**Official Blind Review #1**

**Rating:** 6

**Review:**

This paper focuses on CycleGAN method to show theoretically when the exact solution space is invariant with respect to authomorphisms of the underlying probability spaces for unpaired image-to-image translation.

- The paper provides interesting theoretical results on identifying conditions under which CycleGAN admits nontrivial symmetries and has a natural structure of a principal homogeneous space.  Proposition 2.1 provides interesting insights into the invariance of the kernel space.

- Propositions 2.5 and 2.6 are interesting in that they show that the existence of authomorphisms can worsen the performance of CycleGAN, however, it is unclear that in practice, how could one verify the conditions efficiently before applying CycleGAN.

- The experimental results are interesting, however, they are very limited. Having a toy experiment is a good sanity check, but it would be more interesting to see the performance on a real-world applications, such as medical images or other use-cases brought in the introduction. Also, more discussion on the results provided in Fig 3, confusion matrices, would be very helpful. Are there any intuitions behind the large and low values in the table? It could be interesting to see what are the effects of other parameters such as alpha in producing the results.

- Overall this paper presents interesting results regarding the theory of CycleGANs, however, the numerical results are very limited, and do not justify the motivations discussed in the introduction and the abstract. Moreover, although the paper introduces novel attempts and theoretically analyzing the CycleGAN, the scope of the work seems to be limited, and thus, it does not have a sufficient significance to be published in ICLR. I strongly suggest the authors to expand and provide more experimental evaluations.

** update:
Thanks for your comments! I found the additional experiment useful and better aligned with the purpose of the model. The discussion added clarified the confusion about the automorphism. That is why I decided to change my score.

**Experience Assessment:**

I do not know much about this area.

**Review Assessment: Checking Correctness Of Derivations And Theory:**

I assessed the sensibility of the derivations and theory.

**Review Assessment: Checking Correctness Of Experiments:**

I carefully checked the experiments.

**Review Assessment: Thoroughness In Paper Reading:**

I read the paper at least twice and used my best judgement in assessing the paper.

---

> ### Author Response · Authors · 2019-11-15
> **Thank you for your feedback!**
>
> Thank you very much for your feedback!
>
> As for the MNIST2MNIST task, we observed that adding identity loss here forces the network to preserve the original image quite easily. This is in line with our expectations since the domains are identical and the identity loss should trivially remove the ambiguity. The high values in the confusion matrix correspond to the digit class which is very definitive, and the smaller ones correspond to the cases when the digit class is somewhat ambiguous. It can be for instance digit ‘2’ which is written a bit like ‘6’ with a closed loop in the bottom, and it happens with digits '3', '5', '8' as well.
>
> Following your feedback, we have added some additional experiments on the BRATS 2015 dataset with medical images. In this set of experiments we show how the loss and PSNR (since we have a ground truth available) change when we vary the weight for identity loss, and we compare these values with flipped version of the image. We see that in the absence of identity loss the final CycleGAN loss is very similar for both original and flipped network output, while the PSNR drops significantly for the flipped version. Increasing the identity loss weight does not always result in improved performance in terms of PSNR.
>
> We have  also added additional discussion of some newer image-to-image translation models from the 'automorphism point of view', and we hope that some of the questions we pose can be answered in future work.

---

### Official Review · AnonReviewer3 · 2019-10-24
**Official Blind Review #3**

**Rating:** 8

**Review:**

This is an interesting, timely study.  CycleGAN has attracted a lot of attention in unpaired image-to-image translation. Although the basic idea of CycleGAN seem sensible, its precise behavior is not totally clear--can one really avoid mismatch with CycleGAN? Do we need additional constraints? This paper provides a nice answer the the first question.

Overall I enjoyed reading this paper. The addressed issue is important, the investigation is reasonable, and the results are intuitive and plausible, with clear practical implications. I think it is a good paper.

 I acknowledge I read the authors' response and other reviews and would like to keep my original rating.

**Experience Assessment:**

I have read many papers in this area.

**Review Assessment: Checking Correctness Of Derivations And Theory:**

I carefully checked the derivations and theory.

**Review Assessment: Checking Correctness Of Experiments:**

I carefully checked the experiments.

**Review Assessment: Thoroughness In Paper Reading:**

I read the paper thoroughly.

---

> ### Author Response · Authors · 2019-11-15
> **Thank you for your feedback!**
>
> We kindly thank for your feedback!
>
> We have added some additional experiments on BRATS 2015 dataset to expand the experimental section, and provided an additional discussion of some newer multimodal image-to-image translation models from the 'automorphism point of view'.

---

### Public Comment · ~Tomer_Galanti1 · 2020-03-17
**The Role of Minimal Complexity in Unsupervised Learning of Semantic Mappings**

Thank you for contributing to the theory of unsupervised image to image translation. The existence of automorphisms discussed in Sec. 2.2 was already investigated in "The Role of Minimal Complexity in Unsupervised Learning of Semantic Mappings" by Galanti et al. 2018 https://openreview.net/pdf?id=H1VjBebR- (see Secs. 2 and 4) and in "Generalization Bounds for Unsupervised Cross-Domain Mapping with WGANs" by Galanti et al. 2018 https://arxiv.org/pdf/1807.08501.pdf (see Secs. 4.1 and 4.2). The solution to this problem was further discussed in Sec. 5 of the latter. We would appreciate it if you cite our work.

---

> ### Author Response · Authors · 2020-04-21
> **Thank you for your feedback**
>
> Dear Tomer,
>
> Firstly, thank you very much for your feedback for bringing this work to our attention, it is very interesting. Naturally, we will be adding references to these contributions in the updated version of the paper. Concerning the existence of the automorphisms. The cited sections indeed explain informally that the cycle consistency constraints are not enough to guarantee uniqueness of solutions in general, but the assumptions on the underlying probability space are never fully formalized as far as we can see. Existence of automorphisms in general is a very well-defined property that holds for many (but not all) probability spaces. This assumption, why it matters and how it can be satisfied in practice for domains like natural images and smooth mappings between them is what we wanted to highlight in our submission.

---

### Decision · Program_Chairs · 2019-12-19

**Decision:**

Accept (Poster)

**Comment:**

This paper theoretically studied one of the fundamental issue in CycleGAN (recently gained much attention for image-to-image translation). The authors analyze the space of exact and approximated solutions under automorphisms.

Reviewers mostly agree with theoretical value of the paper. Some concerns on practical values are also raised, e.g., limited or no-surprising experimental results. In overall, I think this is a boarderline paper. But, I am a bit toward acceptance as the theoretical contribution is solid, and potentially beneficial to many future works on unpaired image-to-image translation.